# Co-Creation of Breast Cancer Risk Communication Tools and an Assessment of Risk Factor Awareness: A Qualitative Study of Patients and the Public in India

**DOI:** 10.3390/cancers15112973

**Published:** 2023-05-30

**Authors:** Divya Pillai, Jyoti Narayan, Aleksandra Gentry-Maharaj, Suryanarayana Deo, Dehannathparambil Kottarathil Vijaykumar, Poulome Mukherjee, Nitya Wadhwa, Aparajita Bhasin, Ashutosh Mishra, Anupama Rajanbabu, Ravi Kannan, Zakir Husain, Avinash Kumar, Antonis C. Antoniou, Ranjit Manchanda, Usha Menon

**Affiliations:** 1Translational Health Science & Technology Institute, NCR Biotech Science Cluster, Faridabad—Gurgaon Expressway, Faridabad 121001, India; 2Quicksand Design Studio, New Delhi 110047, India; 3MRC Clinical Trials Unit at UCL, Institute of Clinical Trials & Methodology, Faculty of Population Health Sciences, University College London, London WC1V 6LJ, UK; 4All India Institute of Medical Sciences, New Delhi 110029, India; 5Amrita Institute of Medical Sciences & Research Centre, Kochi 682041, India; 6Cachar Cancer Hospital & Research Centre, Meherpur, Silchar 788015, India; 786/1 College Street, Economics Department, Presidency University Kolkata, Kolkata 700073, India; 8Centre for Cancer Genetic Epidemiology, Department of Public Health and Primary Care, University of Cambridge, Cambridge CB1 8RN, UK; 9Wolfson Institute of Population Health, Barts CRUK Cancer Centre, Queen Mary University of London, Charterhouse Square, London EC1M 6BQ, UK; 10Department of Gynaecological Oncology, Barts Health NH Trust, London EC1A 7BE, UK; 11Department of Health Services Research and Policy, London School of Hygiene & Tropical Medicine, London WC1H 9SH, UK

**Keywords:** breast cancer, risk communication, patient and public involvement, transmedia tools, risk representation, LMIC

## Abstract

**Simple Summary:**

The incidence of breast cancer (BC) is increasing worldwide, and India reported 179,790 cases in 2020. It is important to inform people of risk factors and methods for risk management through interactive risk communication techniques. Affordable, easy-to-understand transmedia tools for the communication of BC risk were co-created by a multidisciplinary team of doctors, clinical researchers, epidemiologists, health economists, digital designers, and public representatives. Qualitative in-depth interviews were conducted in regional languages to assess the views of patients, relatives, the public, and health professionals in India of these prototypes. There was low awareness of BC, with some understanding of age and hereditary risk factors but limited knowledge of reproductive factors amongst the general public, patients, and relatives. Participants favored storytelling techniques (animation and comic strips/infographics) to explain complex issues such as genetic risk and testing. Co-created BC risk communication transmedia tools should be used to support informed decision making.

**Abstract:**

Background: Low awareness of BC and its associated risk factors causes delays in diagnosis and impacts survival. It is critical to communicate BC risk to patients in a format that they are easily able to understand. Our study aim was to develop easy-to-follow transmedia prototypes to communicate BC risk and evaluate user preferences, alongside exploring awareness of BC and its risk factors. Methods: Prototypes of transmedia tools for risk communication were developed with multidisciplinary input. A qualitative in-depth online interview study was undertaken using a pre-defined topic guide of BC patients (7), their relatives (6), the general public (6), and health professionals (6). Interviews were analyzed using a thematic approach. Findings: Most participants preferred pictographic representations (frequency format) of lifetime risk and risk factors and storytelling using short animations and comic strips (infographics) for communicating genetic risk and testing: “In a short time, they explained it very well, and I liked it”. Suggestions included minimizing technical terminology, decreasing the delivery speed, “two-way dialogue”, and using local “language for different locations”. There was low awareness of BC, with some understanding of age and hereditary risk factors but limited knowledge of reproductive factors. Interpretation: Our findings support use of multiple context-specific multimedia tools in communicating cancer risk in an easy-to-understand way. The preference for storytelling using animations and infographics is a novel finding and should be more widely explored.

## 1. Introduction

Breast cancer (BC) is the most common cancer in women worldwide, with a lifetime risk of one in eight [1]. In 2020, there were 19.3 million new BC cases and 10 million deaths [2]. BC incidence is rising across the globe [3]. This is true in India too, where 179,790 BC cases were reported in 2020 [4]. 

Key contributors to the increased incidence of BC are increasing life expectancies and changes in lifestyle (obesity, physical inactivity, tobacco, and alcohol) and reproductive health practices (parity, oral contraceptive use, breastfeeding, hormone replacement therapy) [5]. Genetic predisposition is another key risk factor [6]. Women with the BRCA1 or BRCA2 gene mutations have a lifetime BC risk of 65% to 72% [7,8] and moderate- and low-risk genetic variants are increasingly being identified [9,10]. The lack of awareness of BC and its risk factors contributes to delays in patient presentation and diagnosis at an advanced stage [11,12,13,14]. Creating awareness is crucial if we are to implement risk management strategies and improve disease outcomes [15,16].

The first step to increasing BC awareness is communicating the risk of developing the disease and introducing risk factors to the general public. Risk communication is an interactive process involving the exchange of information and opinions between healthcare professionals and the general public. It includes providing information about and estimates of the incidence and nature of risks and arrangements for risk management [17]. The objective is to modify behavior and allow at-risk groups, patients, and their families to make informed decisions.

In practice, risk communication is a challenge [18]. Basic competence in numeracy is required to understand health statistics, even among highly educated adults [19]. To help patients overcome a fear of numbers and statistical concepts, there is a need to use appropriate methods to communicate medical risks. Many studies have explored visual representation [20,21]. Identifying the appropriate format based on the objective of the communication is crucial. In addition, there are the patient’s own health beliefs related to diseases of concern and trust in the medium whereby risk is communicated [18].

We report on a study where we created BC risk communication prototypes and evaluated prototype preferences amongst patients, relatives, the public, and health professionals in India. Additionally, we explored awareness of BC and its risk factors.

## 2. Materials and Methods

A qualitative, in-depth online interview study was conducted across three Indian states—Kerala (South), Assam (East), and Delhi (North). The multi-disciplinary study team included clinicians, public health researchers, epidemiologists, health economists, and researchers from a design studio with expertise in user-centered innovations in digital design. Institutional Ethics Committee (IEC) approval was obtained from each of the participating sites before commencing the study (CTRI/2020/11/028980). 

### 2.1. Development of BC and Risk Factor Awareness Tools

The prototypes of the transmedia risk communication tools were developed using a stepwise approach. The design team held multiple consultations with study clinicians to gain insights about the disease, and about the target population and their health literacy. The prototypes were built using digital data collection tools based on the narrative-building framework. The process involved streamlining the concepts and content into simple, affordable tools that were more aligned with the understanding of the general public.

Four context-specific BC risk communication tools were developed. The first was to communicate the lifetime risk of developing BC; the options included semicircles (Figure 1a), pie diagrams (Figure 1b), and pictographs (Figure 1c). The cumulative risk of developing BC among Indian females is 1 in 29 [18]. To make it more comprehensible to the public, we converted this to an approximate number of 4 in 100 women who develop BC during their lifetime. The focus of the second prototype was on various BC risk factors (Figure 2) using different pictorial formats. These were developed in English and then translated into Indian languages—Malayalam, Hindi, Assamese, and Bengali. The third and fourth prototypes were in English; they introduced the concept of genes, mutations, gene testing, and the implications (societal and personal) of detecting “faulty” BC genes. They included an animation (Figure 3) and an infographic strip (Figure 4). There were multiple discussions about the color and size of the prototypes. 

To assess participants’ existing knowledge of BC and its risk factors and to refine the communication tool, we conducted interviews of four groups of end users of the tool

Group 1: BC patients and survivors. These were women who were undergoing treatment or had completed treatment in the previous year. Group 2: immediate family members or close relatives of BC patients. Participants were identified by clinicians at each site, ensuring that emotionally vulnerable patients or relatives were not approached. Group 3: unaffected general public. This sample included adult women with no restrictions on age, as the aim of the tool was to increase awareness of BC risk factors and anyone may have female relatives/friends/colleagues who are at risk. It is also important to note that the age distribution for female BC in the Indian population is different from Western populations. In Indian women, the incidence begins to rise from age 25, which is around 10 years lower than in Western populations [22,23]. The public representatives (PRs) were identified by local teams from the community in the vicinity of the participating hospital and from those attending the hospital as caregivers for patients with diseases other than cancer. Additionally, in Delhi, a local patient support group helped to identify PRs from the community. All had no personal history of cancer and no first-degree relatives with cancer.

Group 4: health professionals (HPs) who work with BC patients or are involved in health promotion, as they are likely to use the tool. This included nurses, counsellors, and oncologists working in oncology clinics at the participating hospitals. Written informed consent was obtained from all participants by the site teams.

All interview participants were aged 18 years or older. A sample size of around six per group was considered adequate to provide enough informational power.

In-depth individual interviews were conducted jointly by a team of two researchers, one from public health (DP) and the other from the design team (JN), using a topic guide (Appendix A). The topic guide had six components: knowledge of BC, the experience of sharing a BC diagnosis, awareness of BC risk factors, preference with regards to the risk communication prototypes and suggestions for improvements, willingness to participate in research, and preferred avenues for seeking health information. The interviews, which lasted about an hour, were conducted using an online platform (Go-To meeting) in view of the COVID-19 pandemic. All participants were offered access to the computer/internet facilities at the participating sites, with a local team member providing technical support. The interviews were undertaken in the participant’s preferred language (English, Hindi, Malayalam, Bengali, or Assamese). Where the interviewers were not proficient in the local language (Assamese/Bengali), an onsite research team member was trained to help with the interview process. The site’s clinical staff were available to address any concerns that could arise as a result of the interview.

The interviews were audio-recorded. The recording was only commenced after introductions, so that personal identifying information (names, places, or visuals) was not captured. In addition, no other identifiers, such as the GPS location or IP address of the participants, were captured. The recordings were stored in a secure place at the coordinating site and transcribed and translated into English by a professional translation agency.

### 2.2. Analysis

The participant profiles were described using descriptive statistics. The monthly household income was categorized using the Kuppuswamy Socio-Economic Status scale 2020 [24]. This is a socio-economic scale that has been validated for the Indian population and includes the monthly household income, employment, and educational qualifications. Interviews were undertaken until data saturation was reached. The interviews were analyzed using a qualitative thematic analysis using Excel. Two reviewers coded each interview using an inductive approach (J.N., D.P.). Disagreements were resolved by discussion with a third reviewer (A.B.). The agreed codes were grouped into categories. From these categories, basic and global themes were identified using the framework of the thematic analysis of qualitative data. This reflected the views/experiences of participants, rather than those pre-determined by the researchers.

## 3. Results

The interviews were conducted between October and December 2020. A total of 25 participants were recruited from the three sites—Kerala (8) Delhi (9), and Assam (8). They consisted of seven patients, six relatives, six PRs, and six HPs (Table 1).

The majority were female (76%), with males restricted to relatives and HP. The relatives comprised the spouses (4), a brother (1), and a daughter (1) of BC patients. The median age of those interviewed was 40 (IQR 16.5), and most were college educated (76%). The monthly household income ranged from INR 4000 (equivalent to USD 52) to more than INR 75,000 (equivalent to USD 1011) (Table 2).

From the responses received from the participants based on the topic guide, the following basic and global themes were identified using the framework of the thematic analysis of qualitative data (Table 3).

### 3.1. Varying Awareness Levels of BC and Its Diagnosis

Most patients and relatives were not aware of BC and had a number of misconceptions before the diagnosis; their awareness increased over time by reading online materials and watching videos. A patient admitted that “There was, not much awareness at that time… We had a lot of wrong ideas too, and that’s why I delayed consultation with the doctor… we also searched Google”. This was echoed by others: “ I was not aware of that much till I was diagnosed when we read different articles about it and saw different videos related to it”.

The HPs highlighted a lack of knowledge among older women. “The elderly population don’t have much idea about the disease... The age group of 40 to 50 years knows about this and they are aware of it. But the aged mothers are not aware of it …. They would ask why has this had happened to me”.

Awareness of BC varied among the general public. Most knew little, stating “I have heard vaguely about it…”. Others had come across it: “A girl I know had it. She was told that they would have to operate on her and remove the lump”. The awareness of symptoms was low. Respondents identified increased breast size, weight loss, and fever as symptoms of BC.

Some answers illustrated critical knowledge gaps, for example with regard to cancer sites—“it may occur in the stomach” or “in the place where they get their periods”. Only a few understood the importance of regular screening and seeking prompt medical attention.

### 3.2. Varying Awareness Levels and Some Misconceptions of Breast Cancer Risk Factors

Respondents from the first three groups recognized that BC was a disease affecting women, with the risk increasing with age. “…women are more prone to getting it after a certain age, which is why they are advised to get tested regularly after a certain age”. However, most had a limited understanding of most BC risk factors: “there are many who are not aware of any risk factors” (HP).

Awareness about the hereditary nature of BC was quite high among all groups. The general public knew that BC “can be genetically passed on”. Patients stated that “It could be hereditary… Heard from a doctor on genes…”, and their relatives were also aware of the hereditary nature of BC. This was confirmed by HPs, who stated that “Patients, at most, are aware that it is hereditary”. Awareness levels were high among university graduates and younger individuals. However, most respondents, including university graduates, did not know of specific high-risk genes such as BRCA.

Some answers highlighted a greater lack of knowledge. The only reference to reproductive risk factors was made by one patient who stated, “Females who are single and who don’t have kids are at a higher risk than married females with kids.” The misconception that BC was a communicable disease was a serious erroneous belief. Health professionals stated that there were still some who thought that BC was “a communicable disease. They speak in this manner too, my mother had it, so that’s why I got it as if it were an infectious disease” (HP); “If you sit and eat together, cancer will spread to their body from us.” (public).

With regard to lifestyle factors, there were some comments on exercise. “I never felt that lack of exercise would lead to breast cancer”; “People in the villages don’t know what it is to exercise; only people in the cities do it”. The awareness of the association between lifestyle and cancer was generalized, with all cancers being attributed to smoking and drinking: “many also feel that any type of cancer that happens is due to smoking and drinking only” (HP). Some respondents were aware of excessive smoking and alcohol consumption contributing to the development of lung and liver cancers: “I know that you get lung cancer if you smoke or consume tobacco”, or “Excessive alcohol consumption can cause problems in the liver” (public).

A fairly widespread misconception was that “The men feel, this is female related, it is not meant for us” (HP). However, a few better-educated participants knew that men too could suffer from BC (“heard both men and women can get breast cancer”).

### 3.3. Preference for Risk Communication Prototypes

Risk estimates (Figure 1): three prototypes were developed. Most respondents preferred the pictogram (Figure 1c), as it was “eye-catching” and “easy to understand and self-explanatory,” especially for “people who are not very well-read”. Some suggestions include placing the “four human figures in red in the first line …. (as) they are a little bit scattered here and there.” Only two participants preferred the pie-chart (Figure 1b) and half-circle depictions (Figure 1c) of the lifetime risk of developing BC. Suggestions included changing the color to pink as it “is the color of BC awareness” or the shape to that of the “breast” and relying on “ less of words... more of simplified visuals which anybody can understand”.

Risk factors (Figure 2): four prototypes were developed. Of the 25 participants, 19 preferred a pictographic representation (Figure 2a), though 17 felt that videos would be best. Respondents suggested using human faces instead of drawings and including other risk factors responsible for BC. “There should be a mention of the other risk factor; “With actual human faces, the public might connect better”; “Video representation would be the best”.

Animation about genes: a three-minute English animation (Figure 3) on the basics of genes including high-risk BC genes (BRCA1 and BRCA2) was felt to be informative, comprehensible, and appealing by most patients—“…The animation is excellent and could easily be understood”; “In a short time, they explained it very well, and I liked it”. HPs also appreciated the animation as it helped them to communicate with patients about the diagnosis more easily: “It is a very good video because, when they are explaining, we can understand it nicely too”. However, some respondents felt that the focus on genes might make the animation difficult for people without any idea about genes: “One group of people, I don’t think (will understand it)... because they don’t have any idea about genes”. It was felt that this group might find it easier to understand the hereditary aspects of BC. An HP suggested making two videos catering to different segments of the population. The need to use local languages was commonly expressed: “Video to be made in the regional languages to reach more people…”. Respondents emphasized the need to minimize technical terminology and decrease the delivery speed: “The animation was fast and needs to slow down a bit”. It was suggested that “a two-way dialogue” could improve engagement.

Infographic strip on genetic testing: the storytelling method in the infographic strip (Figure 4) on genetic testing and counselling was received well by all. Modifications such as making “one character a doctor and the other a patient who is going for genetic testing” (HP, patient); using local “language for different locations”, “same dialogue to make a video, not a cartoon…” (HP), and using “the animation and illustration... together” (HP) were put forward.

### 3.4. Other Suggestions

Participants, irrespective of their group, were keen that information on prevention, screening, and counselling was included. “There should also be some information on prevention…”; “Include prevention, apart from early detection, certain lifestyle changes that would reduce your risk.” “Awareness tool on BC screening methods and further steps if it tests positive during gene testing…” This was supported by the HPs: “what will be done and how fast it will be done …. in a convincing manner.” In addition, it was suggested that information on support systems was included, to combat social stigma.

## 4. Discussion

### 4.1. Main Findings

As far as we are aware, our study is the first to highlight a preference for animation and comic strip infographics for risk communication in a low-middle-income country. The prototypes developed are original and add a new dimension to the risk communication literature. The need to include region-specific risk management information was universally emphasized. The findings are of particular relevance to low- and low-middle-income countries. While the specific tools apply to BC, the principles can be adopted for other forms of health risk communication.

In keeping with the literature, we found limited awareness of BC and its risk factors both among the general public and in BC patients and their relatives. Respondents perceived BC to be a hereditary disease that primarily affects women, particularly older women. The knowledge seemed to be linked to the education level of the respondent. There was hardly any awareness of reproductive or lifestyle risk factors associated with BC. An important misconception voiced by some was that BC is a communicable disease.

### 4.2. In the Context of Other Literature

Our participants strongly supported the use of simple communication tools (illustrations, pictograms, animations, and infographics). For risk communication, respondents preferred pictograms with life-form figures (frequency format) to graphs. This is in keeping with the emerging effective evidence-based methods of communicating probabilities [25]. Patients may have a more accurate perception of risk if probabilities are presented as event rates out of groups of 100, 1000, or 10,000 (also called natural frequency formats) [18]. A pictorial display of risk (human or rectangular forms) has been reported to be associated with a better understanding of breast cancer risk when compared to a bar graph in randomized controlled trials [26,27]. In a recent study from Japan on BC risk, it was found that, even among university students, the preference was for pictorial presentation [28,29]. This preference was also noted in cardiovascular risk communication studies [30] and for communicating treatment risk and benefit information to people with different educational and socio-economic backgrounds [20].

Our participants chose a storytelling approach that uses animation and infographic strips to communicate complex information about how risk factors contribute to the development of BC and how this risk can be managed. The co-creation effort resulted in useful suggestions for improvement, such as the use of local languages and reducing the speed of the animation. A study among Italian-speaking women aged 18–30 years reported a more positive effect on BC awareness of videos compared to infographics [31]. Video-based education has previously been shown to significantly improve decision-making abilities concerning treatment options in BC patients [32].

Our findings are consistent with the recently recommended general principles for evidence communication: inform, not persuade; address all the questions and concerns of the target population; offer balance, not false balance, presenting potential benefits and possible harms fairly; disclose uncertainties, being open about a range of possible outcomes; highlight the quality and relevance of the underlying evidence; anticipate misunderstandings and pre-emptively debunk or explain; and use a carefully designed layout in a clear order with sources highlighted [33].

Our findings with respect to low levels of awareness about BC and its risk factors are consistent with previously published reports from India [34,35] and other LMICs such as Nigeria, Kenya, Pakistan, Indonesia, Egypt, and Uganda [36,37,38,39,40,41]. This was confirmed in a recent meta-analysis of 92 studies, which found that, while 84% had heard of BC, only 40% were aware of its risk factors. Similarly to our study, the cohorts in this meta-analysis included healthy women, health professionals, BC patients, and first-degree relatives of BC patients [42]. The 29 most recent (2015–2020) studies included were from China, India, Indonesia, Malaysia, Iran, UAE, Nepal, Pakistan, Cameroon, Ethiopia, Uganda, Central Africa, Zimbabwe, Australia, and Poland.

We found limited awareness of reproductive- or lifestyle-related risk factors. This is in line with studies from low-, middle- and high-income countries, which have reported a lack of knowledge of the risks associated with reproductive practices [35,41,43,44] but an understanding of the hereditary nature of BC [42,43,44,45,46]. The meta-analysis also reported that low awareness about BC reproductive risk factors contrasted with better knowledge about the hereditary nature of BC [42]. High education levels and asset indices were associated with higher levels of BC awareness [34,41], in keeping with our findings. Despite the launch in 2010 of a National Programme for the Prevention and Control of Cancer, Diabetes, Cardiovascular Diseases, and Stroke (NPCDCS), India does not have an organized national BC screening program [47]. This very likely contributes to the limited awareness of the disease.

### 4.3. Strengths and Limitations

One of the key strengths of the study was the use of multiple risk communication formats and the multidisciplinary nature of the team, which involved oncologists, public health researchers, and digital media professionals from both India and the UK. It ensured that scientific concepts were translated into real-time education models that were comprehensible to the general population. Additionally, the involvement of patients, their families, and the general public, the use of local languages for interviews, and the involvement of multiple centers from three of four geospatially distributed regions across India, including Delhi (North), Kerala (South), and Assam (East), ensured the generalizability of our conclusions.

A limitation was the limited sample size, given the breadth and diversity of the Indian population. We were not able to include a site (Mumbai) from Western India, due to delays in securing ethical approval as COVID studies were being prioritized during the pandemic. Nevertheless, within our sample size of 25, the interviews reached saturation. Another limitation is that we used convenience sampling to select participants. While our sample was broadly representative of the Indian socio-economic groupings (measured using the Kuppuswamy scale 24), there was an overrepresentation of university-educated respondents. It is likely that differences in computer literacy contributed to this bias, despite our efforts to minimize this by providing access to the computer/internet facility at the participating sites, with a local team member providing technical support. Finally, interviews were conducted online due to the restrictions imposed by the pandemic. However, local staff facilitated the exchange, thereby decreasing bias related to digital literacy.

There was significant variation in levels of awareness of BC and associated risk factors. It was not possible to ascertain from our study design whether certain sociodemographic populations may have larger knowledge gaps. Different subgroups or people may require or desire different levels of information. These gaps need to be addressed in future research.

## 5. Conclusions

We found that pictorial representation (frequency format) and storytelling using animation and comic strips (infographics) were the preferred options for communicating risk information about BC in India. This is in keeping with observations worldwide. As far as we are aware, our study is the first to highlight a preference for animation and comic strip infographics for risk communication in a low-middle-income country. Awareness of BC and its associated risk factors continues to be low in India, among BC patients, their families, and the general public. Public awareness programs should explore the use of context-specific multimedia tools to provide healthcare information in an easy-to-understand way.

## Figures and Tables

**Figure 1 cancers-15-02973-f001:**
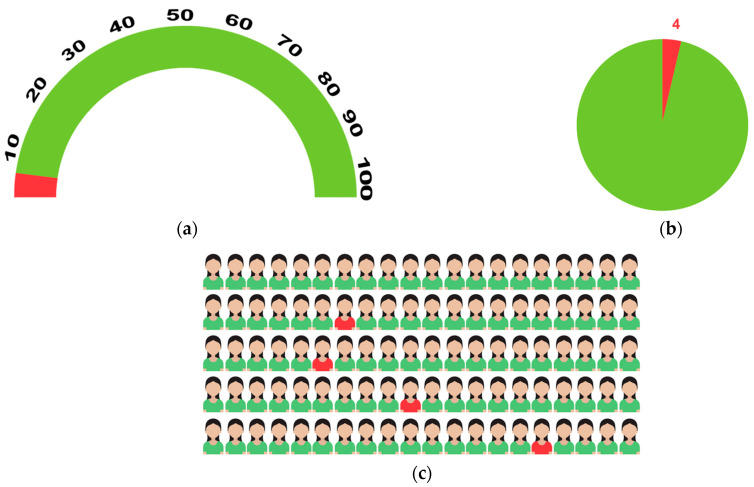
Prototypes depicting lifetime risk for BC. (**a**) In India, 4 in 100 women develop BC during their lifetime, (**b**) In India, 4 in 100 women develop BC during their lifetime, (**c**) In India, 4 in 100 women develop BC during their lifetime.

**Figure 2 cancers-15-02973-f002:**
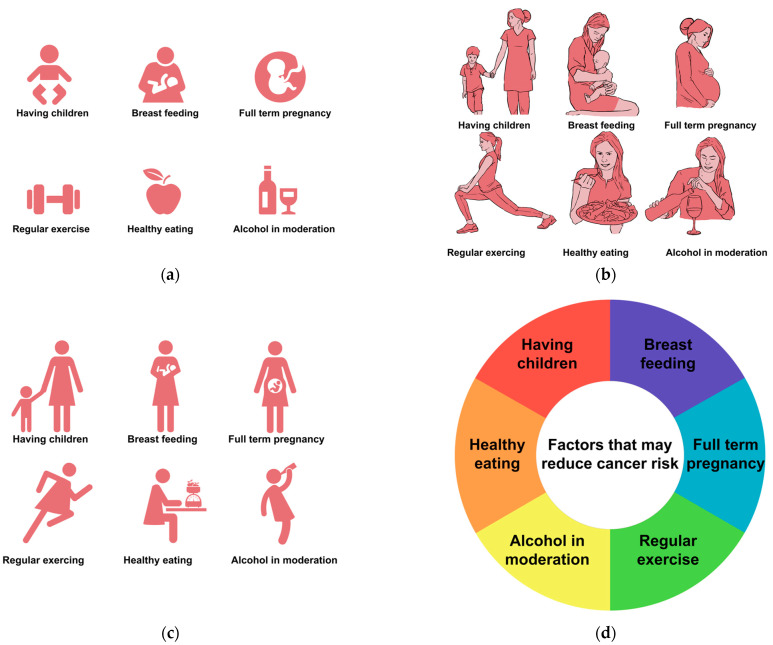
Prototypes depicting breast cancer risk factors. (**a**) Factors that may reduce cancer risk, (**b**) Factors that may reduce cancer risk, (**c**) Factors that may reduce cancer risk, (**d**) Factors that may reduce cancer risk.

**Figure 3 cancers-15-02973-f003:**
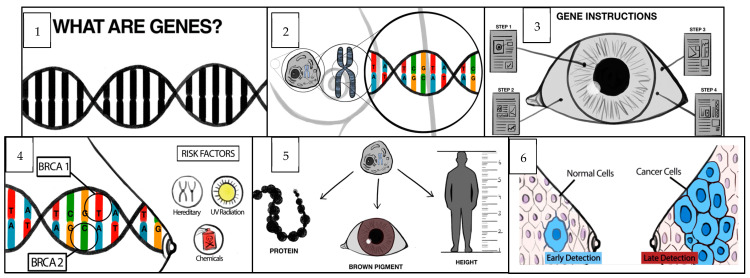
Animation depicting the role of genes in BC evolution.

**Figure 4 cancers-15-02973-f004:**
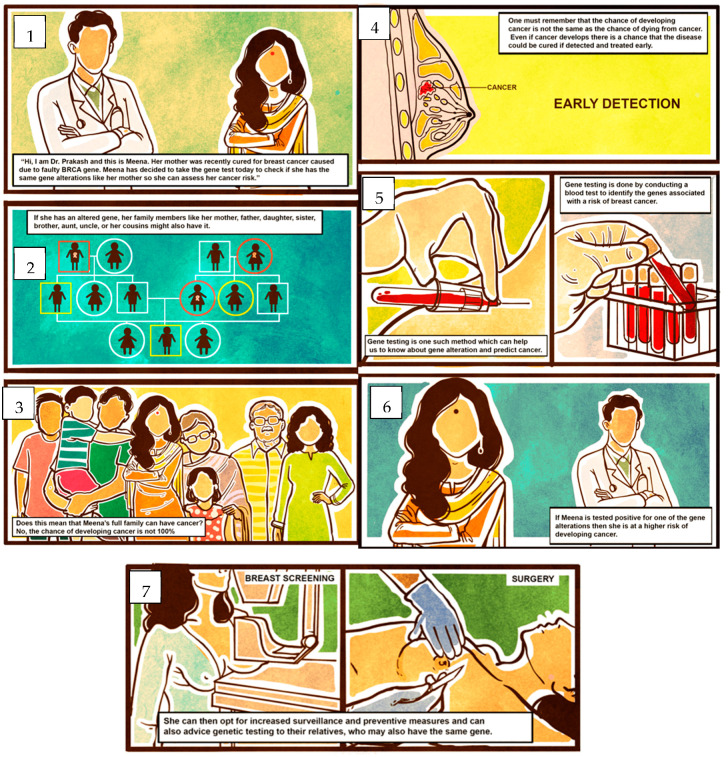
Infographic strip on genetic testing for BC.

**Table 1 cancers-15-02973-t001:** Demographic details of the participants.

S. No.	Participant Category	Age	Gender	Qualification	Employment	Marital Status	Monthly Household Income	Kuppuswamy Score
1.	BC Patient 1	40	Female	Graduate	Not employed	Married	75,000	Lower middle (III)
2.	BC Patient 2	37	Female	Postgraduate	Teacher	Married	75,000	Upper middle (II)
3.	BC Patient 3	41	Female	Postgraduate	Employed	Single	150,000	Upper (I)
4.	BC Patient 4	41	Female	Postgraduate	Employed	Married	100,000	Upper (I)
5.	BC Patient 5	50	Female	Postgraduate	Chartered accountant	Married	300,000	Upper (I)
6.	BC Patient 6	50	Female	Intermediate (12)	Not employed	Married	15,000	Upper lower (IV)
7.	BC Patient 7	31	Female	High school (9)	Not employed	Widow	0	Upper lower (IV)
8.	Relative to BC 1	25	Female	Postgraduate	Student	Single	25,000	Upper lower (IV)
9.	Relative to BC 2	77	Male	Postgraduate	Retired	Married	15,000	Lower middle (III)
10.	Relative to BC 3	34	Male	High school (10)	Driver	Married	15,000	Lower middle (III)
11.	Relative to BC 4	27	Male	High school (10)	Textile shop worker	Single	4000	Upper lower (IV)
12.	Relative to BC 5	44	Male	Postgraduate	Architecture	Married	180,000	Upper (I)
13.	Relative to BC 6	55	Male	Postgraduate	Chartered accountant	Married	300,000	Upper (I)
14.	Lay public 1	30	Female	Postgraduate	Public policy researcher	Single	300,000	Upper (I)
15.	Lay public 2	21	Female	No Formal Education	House help	Single	13,000	Upper lower (IV)
16.	Lay public 3	44	Female	Postgraduate	Not employed	Married	100,000	Upper middle (II)
17.	Lay public 4	49	Female	Graduate	Coordinator	Married	40,000	Upper middle (II)
18.	Lay public 5	42	Female	Middle School	Social worker	Married	10,000	Lower middle (III)
19.	Lay public 6	21	Female	Postgraduate	Student	Single	120,000	Upper middle (II)
20.	Health Provider 1	32	Female	Graduate	Nurse	Married	130,000	Upper middle (II)
21.	Health Provider 2	34	Female	Graduate	Nurse	Single	80,000	Upper middle (II)
22.	Health Provider 3	22	Female	Graduate	Physician assistant	Single	20,000	Upper middle (II)
23.	Health Provider 4	34	Female	Graduate	Nursing assistant	Married	50,000	Upper middle (II)
24.	Health Provider 5	48	Male	Postgraduate	Radiologist	Single	100,000	Upper (I)
25.	Health Provider 6	46	Female	Postgraduate	Pathologist	Married	250,000	Upper (I)

**Table 2 cancers-15-02973-t002:** Summary of key demographic details by participant category.

Participant Category	No. of Participants	Median Age (Min.–Max)	Median Income (Min.–Max)	Kuppuswamy Socio-Economic Status Scale 2020			
				Upper (I)	Upper Middle (II)	Lower Middle (III)	Upper Lower (IV)
BC patients	7	41 (31–50)	75,000 (0–300,000)	43	14	14	29
Relatives	6	44 (25–77)	20,000 (4000–300,000)	33	0	33	33
General public	6	35 (21–49)	70,000 (10,000–300,000)	17	50	17	17
Health professionals	6	36 (22–48)	90,000 (20,000–300,000)	33	50	17	0

$1 = ₹75 (≈); £1 = ₹103 (≈).

**Table 3 cancers-15-02973-t003:** Main themes that emerged.

Breast cancer and its diagnosis—varying levels of awareness	Lack of awareness of symptoms
	Limited awareness of BC among the general public
	Poor knowledge among older women
	Misconceptions associated with diagnosis
	Limited awareness of patients and relatives of BC before the diagnosis
Breast cancer risk factors —varying levels of awareness and some misconceptions	Limited understanding and knowledge of BC risk factors
	Misconception that BC was a disease affecting women only
	Awareness that age was associated with increasing BC risk
	Some awareness about the hereditary nature of BC but no in-depth knowledge of genetic abnormalities/ BRCA pathogenic variants
	Higher awareness levels among graduates and younger generations
	Awareness that smoking and alcohol consumption were risk factors for lung and liver cancers but not breast cancer
	Misconception that BC was a communicable disease

## Data Availability

The study involved qualitative research. The blinded audio recordings in regional languages were transcribed and translated into English by an external translation agency.

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
