# Peer review of "Co-Creation of Breast Cancer Risk Communication Tools and an Assessment of Risk Factor Awareness: A Qualitative Study of Patients and the Public in India"

_cancers, 2023, doi:10.3390/cancers15112973_

Round 1

Reviewer 1 Report

This paper reports a qualitative study of breast cancer patients, relatives, lay public and health professionals to assess their views in relation to breast cancer risk communication tools in India. While the study question is important and the methodology appropriate, I have a number of comments and suggestions which I believe will improve the paper. At this point I recommend a revision.

·       The title needs to include the study design e.g. “qualitative” and setting e.g. “India”. This is not really a pilot study.

·       The Abstract could use with further details around the aims and setting. Also the first  sentence of the Findings section should be in the Methods.

·       Is there any publications from the design of the communication approach or reference to support these/their development further?

·       In the Methods, more information on the sample in terms of the 4 different groups and the size of the sample would be helpful. Why did you choose these groups? Why did the public sample include women of all ages and not targeted at women of breast screening age? A bit more information around the reasons for sampling would be helpful. Also, the interview schedule or questions – could this be added as a supplementary file. How long did interviews last? Reference to thematic analysis needed and further information about saturation.

·       Table 1 should be In the Results and not the Methods.

·       The themes you present sound more like a list of topics/questions that were asked and not necessarily themes that arose from the interviews – could you re-frame these a bit to capture the actual theme/finding that you found?

·       Add identifiers to the quotes in the text would be helpful.

·       I wonder whether having the prototype figures in the Results is appropriate – to me this should be in the Methods or a supplementary file (linked to the Methods) as they are not Results. I see that that you have referred to them in the Methods so please remove in Results and consider having the figures in a supplementary file rather than displayed in the text.

·       Limitation was the small sample size in each of the domains.

·       The main finding of limited awareness of breast cancer risk factors is not new in both high and low/low-middle income countries (as noted in 4.2). The unique part of this study is rather the study setting and information around the prototypes. A bit more of a shift on that in the Discussion is welcomed and what the importance of those findings are for current and future policy and practice.

Author Response

Thank you for undertaking a detailed and thorough review of our manuscript. We have found the comments extremely helpful in improving the manuscript. We would like to extend our appreciation for your time and invaluable suggestions!

Reviewer 2 Report

Thank you for proposing me to review this work. Authors analyze awareness of BC and risk factors in India. They also obtain results on preferences about tools for communicating breast cancer risk. Breast cancer patients, their relatives and the general public is interviewed, as well as health professionals. The article is well structured and its results are in agreement with previous works.

I would like propose to the authors some recommendations, in case they might find them useful.

- Although I believe that the main purpose of the study is to collect preferences and suggestions on communication tools, a not insignificant part of the interviews is focused on knowing what is the knowledge about breast cancer and its risk factors. Perhaps the title of the article could capture this double result (Knowledge of BC and views on communication tools).

- It seems that this is a pilot study prior to another to be conducted in the future, then, could the authors briefly explain what is the ultimate goal of that future study and what practical implications it may have?

- The Indian numbering system may be difficult to understand for people who are not used to them. I would therefore recommend the authors to explain at some point their equivalence to western numbering.

- I think it would be useful for readers to know which variables are incorporated in the Kuppuswamy score included in Table 2 (Income, employment, qualification...?).

- I think it would be appropriate to know the country context if there is a breast cancer screening program in India, at least in some regions, how it works and what participation of women it has. A quick search about this topic (Breast cancer screening existence in India: A nonexisting reality, Singh et al, doi: 10.4103/0971-5851.171539) seems to describe a less than optimistic situation. Perhaps in the discussion they could comment on whether or not this lack has any bearing on the general public's lack of knowledge about breast cancer and its risk factors.

- In my opinion, there could be a bias in the educational level of the interviewees; for example, it has struck me that in the patient, relative and control groups, more than half of the interviewees have Postgraduate Qualification, I think that could be not representative of the general population qualification. The sample may be broadly representative of the Indian socioeconomic group, using the Kuppuswamy score, but I think the authors should emphasize that it may not be as representative with respect to academic qualification, especially since results on level of knowledge and preferences about tools for communicating breast cancer risk may be sensitive to this bias.

- Perhaps this overrepresentation of people with a Postgraduate Qualification is related to the online conduct of the interviews. Could the authors specify (in Materials and Methods) whether these interviews were done individually or in groups and through which platform (Teams, Zoom...)?

- As expected, people with higher qualifications have more information and are more aware of the risk factors for breast cancer. In the Results, it may be useful to separate answers that imply a great lack of knowledge (cancer is a transmissible disease or uncertainty about the anatomical location) from others that, in my opinion, are less serious (that breast cancer only affects women is a fairly widespread opinion in all countries). Are there any sociodemographic characteristics that allow us to identify which type of population has large knowledge gaps about breast cancer? If in the future this study could generate public health policy proposals in India, it might be appropriate to construct different tools for different population groups, as an HP suggested.

Author Response

We appreciate your thorough and insightful review of our manuscript. We went over all of your suggestions and comments and updated the paper accordingly. I appreciate your time and helpful advice.
